# PBRM1 Immunohistochemical Expression Profile Correlates with Histomorphological Features and Endothelial Expression of Tumor Vasculature for Clear Cell Renal Cell Carcinoma

**DOI:** 10.3390/cancers14041062

**Published:** 2022-02-20

**Authors:** Kazuho Saiga, Chisato Ohe, Takashi Yoshida, Haruyuki Ohsugi, Junichi Ikeda, Naho Atsumi, Yuri Noda, Yoshiki Yasukochi, Koichiro Higasa, Hisanori Taniguchi, Hidefumi Kinoshita, Koji Tsuta

**Affiliations:** 1Department of Pathology, Kansai Medical University, 2-3-1 Shin-machi, Hirakata 573-1191, Japan; saigakaz@hirakata.kmu.ac.jp (K.S.); ikedaj@hirakata.kmu.ac.jp (J.I.); naatsumi@hirakata.kmu.ac.jp (N.A.); nodayur@hirakata.kmu.ac.jp (Y.N.); tsutakoj@hirakata.kmu.ac.jp (K.T.); 2Department of Urology and Andrology, Kansai Medical University, 2-3-1 Shin-machi, Hirakata 573-1191, Japan; yoshidtk@takii.kmu.ac.jp (T.Y.); ohsugih@hirakata.kmu.ac.jp (H.O.); taniguhi@hirakata.kmu.ac.jp (H.T.); kinoshih@hirakata.kmu.ac.jp (H.K.); 3Department of Genome Analysis, Institute of Biomedical Science, Kansai Medical University, 2-5-1 Shin-machi, Hirakata 573-1010, Japan; yasukocy@hirakata.kmu.ac.jp (Y.Y.); higasako@hirakata.kmu.ac.jp (K.H.)

**Keywords:** clear cell renal cell carcinoma, histomorphological features, PBRM1, immunohistochemistry, architectural patterns, endothelial cells

## Abstract

**Simple Summary:**

The PBRM1 protein, whose gene is the most frequently mutated one in clear cell renal cell carcinoma (ccRCC) following *von Hippel-Lindau*, has been proposed as a potential biomarker for ccRCC. However, the association of the PBRM1 immunohistochemical expression with histomorphological features of ccRCC and the endothelial expression of tumor vasculature, which is an important role of the tumor microenvironment related to treatment response, is little known. Recently, our research team has established a vascularity-based architectural classification of ccRCC correlated with angiogenesis and immune gene expression signatures, which could provide prognostic information and function as a surrogate for treatment selection. In the present study, we found the PBRM1 expression was correlated with the architectural patterns. Furthermore, we demonstrated that endothelial expression tended to be lost in cases with low PBRM1 expression. This correlation implied the orchestrated expression of PBRM1, raising the possibility that the cancer cells and their microenvironment interact in ccRCC.

**Abstract:**

Loss of the polybromo-1 (PBRM1) protein has been expected as a possible biomarker for clear cell renal cell carcinoma (ccRCC). There is little knowledge about how PBRM1 immunohistochemical expression correlates with the histomorphological features of ccRCC and the endothelial expression of tumor vasculature. The present study evaluates the association of architectural patterns with the PBRM1 expression of cancer cells using a cohort of 425 patients with nonmetastatic ccRCC. Furthermore, we separately assessed the PBRM1 expression of the endothelial cells and evaluated the correlation between the expression of cancer cells and endothelial cells. PBRM1 loss in cancer cells was observed in 148 (34.8%) patients. In the correlation analysis between architectural patterns and PBRM1 expression, macrocyst/microcystic, tubular/acinar, and compact/small nested were positively correlated with PBRM1 expression, whereas alveolar/large nested, thick trabecular/insular, papillary/pseudopapillary, solid sheets, and sarcomatoid/rhabdoid were negatively correlated with PBRM1 expression. PBRM1 expression in vascular endothelial cells correlated with the expression of cancer cells (correlation coefficient = 0.834, *p* < 0.001). PBRM1 loss in both cancer and endothelial cells was associated with a lower recurrence-free survival rate (*p* < 0.001). Our PBRM1 expression profile indicated that PBRM1 expression in both cancer and endothelial cells may be regulated in an orchestrated manner.

## 1. Introduction

Clear cell renal cell carcinoma (ccRCC), the most frequently diagnosed histologic subtype of adult RCC [1], is associated with a hyperangiogenic state due to the overproduction of vascular endothelial growth factor (VEGF) by loss of *von Hippel-Lindau (VHL)* gene function [2]. In addition to targeted therapy for these angiogenesis pathways such as VEGF receptor—tyrosine kinase inhibitors (TKIs) [3], novel systemic immunotherapy agents have improved patient survival in metastatic RCC [4,5]. However, predictive biomarkers for both the prognostic and therapeutic implications of RCC remain lacking in a clinical setting [6].

Recent genomic advances using exome sequencing revealed that the *PBRM 1* gene encoding the protein polybromo-1, which is a subunit of the SWI/SNF chromatin remodeling complex, is a second major ccRCC cancer gene, following the *VHL* gene [7,8]. Several studies have shown that the loss of PBRM1 protein has been confirmed as a possible biomarker for ccRCC, which is associated with adverse pathological factors and poor patient outcomes [9,10]. Subsequently, our research team presented a novel scoring system to predict recurrence after radical surgery using standard pathologic factors incorporating immunohistochemical (IHC) expression of PBRM1 [11]. Furthermore, because *PBRM1* is considered not only a key driver gene of ccRCC but also a key regulator of tumor cell-autonomous immune response in ccRCC, the influence of PBRM1 loss on the response to immune checkpoint inhibitors (ICIs) has been investigated [7,12,13].

Recently, we first demonstrated that histological phenotypes, such as clear or eosinophilic types, were significantly correlated with survival outcomes and response to TKIs and ICIs in patients with ccRCC, which could be applied as a predictive marker for treatment selection [14]. Additionally, we established the vascularity-based architectural classification of ccRCC in accordance with nine architectural patterns, which corresponded to both angiogenesis and immune gene expression signatures [15]. Although the prognostic and therapeutic significance for architectural patterns of ccRCC has been shown [16,17], there is little knowledge on how genomics and subsequent protein expressions are reflected in histomorphological features [18].

To evaluate the association of the PBRM1 expression with histomorphological features, we semiquantitatively re-evaluated the expression by using the PBRM1-stained slides used in our previous study [11]. In addition, we noticed that the expression in vascular endothelial cells, which has been used as one of the internal positive controls in some studies [10,11], tended to decrease or disappear in the PBRM1 loss cases. However, there is little evidence regarding PBRM1 expression of the tumor vasculature, which plays an important role in the tumor microenvironment [19]. In the present study, we aimed to evaluate whether the histomorphological features of ccRCC correlate with the PBRM1 expression of cancer cells. Furthermore, we separately evaluated the PBRM1 expression of the vascular endothelial cells and examined the PBRM1 expression profiles of cancer cells and endothelial cells.

## 2. Materials and Methods

### 2.1. Case Selection

This study was performed under the institutional review board’s approval at Kansai Medical University Hospital (No. 2018109 and No. 2020222). As in our previous report [15], data for 436 patients who underwent extirpative surgery for nonmetastatic ccRCC were identified from the institutional database between 2006 and 2017. Of these, 11 patients were excluded from this study due to an insufficient supply of pathological materials for immunohistochemistry. Thus, 425 cases with nonmetastatic ccRCC (cT1-4N0-1M0) were retrospectively analyzed. Our institutional database of RCC contains pathological findings, which were re-evaluated by a genitourinary pathologist (C.O.) based on the 2016 World Health Organization (WHO) classification [20] and the 2017 TNM staging system [21] as previously described [11,14,15]. All ccRCCs were histologically diagnosed when the carcinoma contained typical ccRCC histology and/or showed diffuse membranous positivity of carbonic anhydrase IX by immunohistochemistry [20]. Pathological prognostic factors, including pathological TNM stage, WHO/International Society of Urological Pathology (WHO/ISUP) grade, and necrosis, were collected [22].

### 2.2. Evaluation of Histomorphological Features

All histomorphological features were evaluated by C.O., blinded to clinical outcomes, using whole-tissue sections of H&E-stained slides. Histological phenotype, based on cytoplasmic features, such as clear, mixed, or eosinophilic, and vascularity-based architectural classification, based on nine architectural patterns, such as compact/small nested, macrocyst/microcystic, tubular/acinar, alveolar/large nested, thick trabecular/insular, papillary/pseudopapillary, solid sheets, and sarcomatoid and rhabdoid, were determined at the highest-grade area as previously described [15].

### 2.3. Tissue Microarray (TMA) Construction and Immunohistochemistry of PBRM1 

As previously described [11,23,24], TMA was constructed from duplicate 2 mm cores of representative tumor locations (including the highest-grade area) in each case. The morphological patterns of each core were also assessed based on the nine architectural patterns included in the vascularity-based architectural classification [15]. A primary antibody against PBRM1 (rabbit polyclonal, dilution 1:200; Atlas Antibodies AB, Bromma, Sweden) was used according to the manufacturer’s protocols of the Ventana Discovery Ultra Autostainer (Roche Diagnostics, Indianapolis, IN, USA). PBRM1 was visualized with OptiView and an amplification kit (Ventana Medical System, Tucson, AZ, USA). The same PBRM1-stained slides from our previous study [11] were used in the present study. The nuclear expression of cancer cells was semiquantitatively assessed, referring to the internal positive controls (inflammatory cells or stromal fibroblasts), using the H-score. The score was determined by multiplying the staining intensity (0, none; 1, weak; 2, moderate; and 3, strong) and the percentage of positive cells (range: 0–300). The final scores (average H-score for the two cores) were determined as previously described [23]: H-score ≤ 20 was considered for PBRM1 loss, and H-score > 20 was considered for PBRM1 retention in cancer cells. An IHC evaluation was performed by two pathologists (K.S. and C.O.), and discordant cases were resolved by consensus. Next, we separately evaluated the nuclear expression of endothelial cells within the tumor area and scored them as follows: 0, none; 1, focal weak; 2, diffuse weak; or 3, diffuse strong. The scores of endothelial cells were finally stratified as PBRM1 loss (score: 0–1) and PBRM1 retention (score: 2–3). The representative PBRM1 expressions of cancer cells and endothelial cells are presented in Figure 1.

### 2.4. Statistical Analysis

Statistical analyses were performed using EZR version 1.54 (Saitama Medical Center, Jichi, Japan) [25]. A two-sided *p* < 0.05 was considered statistically significant. A Chi-squared test for categorical variables was used to evaluate the statistical significance among two or more groups. The t-statistic in linear regression analysis and one-way ANOVA analysis were used to evaluate the statistical significance among the architectural patterns. Interobserver agreement was statistically assessed using kappa statistics. Correlations between the two variables were evaluated using Spearman’s rank correlation test. Recurrence-free survival (RFS; recurrence was calculated on imaging from the date of surgery to the date of recurrence) was assessed using the Kaplan–Meier method with the log-rank test.

## 3. Results

### 3.1. Patients’ Characteristics and PBRM1 Expression in Cancer Cells

The median age of the patients was 65 years (IQR, 56–73 years). The male to female ratio was 2.8:1 (312 males and 113 females). The rate of TNM stage III or IV, WHO/ISUP grade 3 or 4, and the presence of necrosis was 24.0% (102/425), 32.3% (137/425), and 15.3% (65/425), respectively. Of the 425 patients, 57 (13.4%) experienced a recurrence of ccRCC during a median follow-up of 62.6 months (IQR, 33.8–94.0 months).

Cases with PBRM1 loss and PBRM1 retention were observed in 148 (34.8%) and 277 (65.2%) patients, respectively. The interobserver variability showed good agreement between the two pathologists (kappa = 0.84). The PBRM1 expression of clinicopathological factors is shown in Table 1.

### 3.2. Association of PBRM1 Expression in Cancer Cells with Clinicopathological Factors

Loss of PBRM1 expression was significantly associated with worsened pathological prognostic factors, such as TNM stage, WHO/ISUP grade, and the presence of necrosis (all *p* < 0.001; Figure 2A). Regarding the association of PBRM1 expression with histomorphological features, cases with PBRM1 loss were significantly observed in the eosinophilic type, which is related to high gene expression signature scores of effector T-cells, immune checkpoint molecules, and epithelial and mesenchymal transitions [14], among other histologic phenotypes. Similarly, cases with PBRM1 loss were significantly observed in category 3, which is associated with a low gene signature of angiogenesis and high gene signatures of effector T-cell and immune checkpoint [15], among vascularity-based architectural categories (both *p* < 0.001; Figure 2B).

### 3.3. Association of PBRM1 Expression in Cancer Cells with Architectural Patterns

Regarding the association of PBRM1 expression with architectural patterns in the highest-grade area, tumors with PBRM1 loss were observed in 50/177 (28.2%) of compact/small nested, 1/36 (2.8%) in macrocyst/microcystic, 7/63 (11.1%) in tubular/acinar, 20/47 (42.6%) in alveolar/large nested, 37/55 (67.3%) in thick trabecular/insular, 10/20 (50%) in papillary/pseudopapillary, 8/9 (88.9%) in solid sheet, and 15/18 (83.3%) in sarcomatoid/rhabdoid patterns (Table 2). Representative images of PBRM1 expression in each architectural pattern are shown in Figure 3.

To evaluate the correlation between architectural patterns and PBRM1 expression (H-score), multiple linear regression analysis was performed (Figure 4). Macrocyst/microcystic (t statistic = 7.734, *p* < 0.001), tubular/acinar (t statistic = 4.228, *p* < 0.001), and compact/small nested (t statistic = 1.95, *p* = 0.0519) were positively correlated with the PBRM1 expression although compact/small nested was not statistically significant. On the other hand, thick trabecular/insular (t statistic = −5.98, *p* < 0.001), sarcomatoid/rhabdoid (t statistic = −3.829, *p* < 0.001), solid sheets (t statistic = −2.965, *p* = 0.0032), alveolar/large nested (t statistic = −2.935, *p* = 0.0035), and papillary/pseudopapillary (t statistic = −2.016, *p* = 0.0444) were negatively correlated with the PBRM1 expression. 

Of 403 cases where two cores were assessed for PBRM1 expression (22 out of 425 cases were missing one core), 77 (19.1%) showed heterogeneity of PBRM1 expression (H-score ≤ 20 vs. >20) between cores. Therefore, we examined whether PBRM1 expression was correlated with the architectural patterns of the corresponding area by assessing a total of 828 cores. It was revealed that PBRM1 expression was correlated with the architectural patterns among all of the evaluated cores. Notably, this association between PBRM1 expression and the architectural patterns assessed in the highest-grade area, namely, macrocyst/microcystic, tubular/acinar, and compact/small nest, had significantly higher PBRM1 expressions (H-score) compared to the other patterns (*p* < 0.001, *p* < 0.001, and *p* < 0.05, respectively) (Figure 5). 

### 3.4. Association between Cancer Cells and Endothelial Cells

#### 3.4.1. Correlation between PBRM1 Expression in Cancer Cells and Endothelial Cells

A positive correlation between PBRM1 expression in cancer cells and endothelial cells was confirmed (correlation coefficient = 0.834, *p* < 0.001; Figure 6A). 

#### 3.4.2. Prognostic Significance of PBRM1 Expression in Cancer Cells and Endothelial Cells

Survival curve analysis showed that the 5-year RFS rate was significantly lower in patients with PBRM1 loss than in those with PBRM1 retained in cancer cells (71.1% versus 96.1%, *p* < 0.001; Figure 6B). Similarly, the 5-year RFS rate was significantly lower in patients with PBRM1 loss than in those with PBRM1 retained in endothelial cells (72.5 versus 95.6%, *p* < 0.001; Figure 6C).

## 4. Discussion

Typical histological features of ccRCC consist of neoplastic cells with clear cytoplasm and a vascular network of small and thin-walled blood vessels, activated by hypoxia-inducible factors following *VHL* inactivation [20]. Although the most common architectural pattern of ccRCC is compact/small nested with an extensive vascular network, the morphologic intratumoral heterogeneity of ccRCC has been recognized [15,16,17]. Recent findings have shown that *VHL* mono-driver tumors are characterized by low-grade and indolent behavior with minimum intratumoral heterogeneity [26]. In contrast, tumors characterized by high-grade and aggressive behavior include multiple clonal drivers that exhibit truncal aberrations of ccRCC epigenetic-related genes: the SWI/SNF chromatin remodeling complex gene *PBRM1*, histone deubiquitinate gene *BAP1,* and histone methyltransferase gene *SETD2* [8,26]. Högner et al. also showed that the combined loss of PBRM1 and VHL may contribute to tumor aggressiveness [27]. However, little is known about the ways these genetic abnormalities impact the histomorphological features of ccRCC.

In the current study, we provided several insights into the PBRM1 IHC expression profile of ccRCC. First, we revealed the association of PBRM1 expression with histological phenotype based on cytoplasmic features [14] and vascularity-based architectural classification [15] (Figure 2B), both of which stratify patient prognosis. For histological phenotype, the eosinophilic type was significantly correlated with PBRM1 loss, followed by mixed type, whereas for vascularity-based architectural classification, category 3 was significantly enriched in the PBRM1 loss group, followed by category 2. These results indicated that PBRM1 loss was correlated with novel poor prognostic factors based on histomorphological features. Consistent with the previous reports [28,29,30,31], we showed the adverse prognostic factors of ccRCC, such as high TNM stage and WHO/ISUP grade or presence of necrosis, were significantly associated with PBRM1 loss (Figure 2A). While a study of localized RCC using TMA failed to show the prognostic role of PBRM1 loss after adjusting for the significant prognostic clinicopathological parameters [32], multivariable models of our prior study showed that PBRM1-negativity is an independent prognostic factor for RFS [11]. Thus, our findings suggest that the additional epigenetic change increases the aggressiveness of ccRCC and results in a poor prognosis. 

Second, we showed architectural patterns based on a vascularity-based architectural classification [15], assessed in the highest-grade area in 425 nonmetastatic ccRCC, correlated with the PBRM1 expression profile. Macrocyst/microcystic, tubular/acinar, and compact/small nested patterns characterized by enrichment of the vascular network (corresponded to category 1) were positively correlated with PBRM1 expression, whereas alveolar/large nested, thick trabecular/insular, and papillary/pseudopapillary patterns characterized by the widely spaced-out vascular network (corresponded to category 2), or solid sheets and sarcomatoid/rhabdoid patterns characterized by scattered vascularity without a vascular network (corresponded to category 3) were negatively correlated with the PBRM1 expression (Figure 4). These results indicate that PBRM1 expression patterns differ among the architectural patterns of ccRCC with or without an extensive vascular network.

Third, in the evaluation of 828 cores considering intratumor heterogeneity, we also demonstrated architectural patterns in macrocyst/microcystic, tubular/acinar, and compact/small nested associated with significantly higher PBRM1 expression (H-score) compared to the other patterns (Figure 5), which suggested that PBRM1 expression profile correlated well with the ccRCC architectural patterns, even with intratumoral heterogeneity. Although intratumoral heterogeneity of ccRCC has been reported based on DNA sequencing and chromosome aberration analysis [33], we showed that loss of PBRM1 protein reflects morphologic heterogeneity and aggressive architectural patterns of ccRCC.

The role of PBRM1 protein expression for clinical decisions is not only being a biomarker of prognostic prediction but also providing information on molecular mechanisms and potential therapeutic targets. In the present study, we showed the prognostic predictive ability of PBRM1 loss in nonmetastatic ccRCC, while Cai et al. also showed that PBRM1 could improve the predictive accuracy for survival outcomes of metastatic RCC patients treated with tyrosine kinase inhibitors (TKIs) [34]. Recently, the effectiveness of systemic therapies (TKIs vs. ICIs) in patients with the *P**BRM1* mutation status of ccRCC has also been investigated [12,13,35,36,37]. Although some studies have shown that patients with PBRM1 loss in ccRCC experience increased clinical benefit from ICIs [12,35], data on the effect of PBRM1 loss regarding immune responsiveness are inconsistent [13,36,37]. According to our previous study, category 3 of the vascularity-based architectural classification, which is related to loss of PBRM1 expression, was significantly associated with an inflamed and excluded immunophenotype in the localized ccRCC cohort and significantly enriched in effector-T cell and immune checkpoint gene signatures in the TCGA-KIRC cohort [15]. We have also shown that in ccRCC, including eosinophilic features related to loss of PBRM1 expression, significant clinical benefit was observed in the ICI therapy group compared to the TKI therapy group (*p* = 0.035) [14].

Contrary to our findings, however, some studies showed that *PBRM1* mutations were associated with increased angiogenesis, decreased immune infiltrates, and poor response to ICIs [13,37]. While these controversial findings have yet to be resolved, the *PBRM1* mutation does not directly determine the loss of the corresponding protein or function [38]. Because some discrepancies between *PBRM1* mutation and PBRM1 IHC expression have been reported, a comprehensive investigation, including *PBRM1* mutation, PBRM1 expression, and histomorphological features, should be conducted. Recently, Lin et al. evaluated the influence of PBRM1 loss for treatment response, focusing on the “immunogenic” tumor microenvironment [13]. However, the “non-immunogenic” tumor microenvironment, including endothelial cells, is also an important factor for appropriate treatment strategies because combined therapies of TKIs and ICIs have been applied for metastatic ccRCC [19,39]. Nevertheless, there are a few studies focusing on the expression of PBRM1 in endothelial cells of ccRCC.

To the best of our knowledge, we are the first to have demonstrated that the PBRM1 IHC expression of endothelial cells is correlated with the expression of cancer cells, which suggests that the vascular endothelial cells may also be genetically or immunohistochemically abnormal (Figure 6). Although we should consider the possibility of a marked reduction in the protein expression due to insufficient or unequal fixation [40], positive expression of internal control such as inflammatory cells or stromal fibroblasts was confirmed in the present study (Figure 1). Angiogenesis also plays a central role in ccRCC tumorigenesis and progression, regulating the immune landscape through abnormal tumor vessel formation [39]. Our observation showed that the tumor vasculature among the vascularity-based architectural pattern of category 1 vs. categories 2 and 3 was different. The specific mechanism underlying the association of decreased PBRM1 expression with the architectural patterns without a vascular network is still unclear, but the interaction of cancer cells and endothelial cells may be suggested. In the current treatment strategies, including angiogenic therapy, the understanding of the epigenetic abnormality between cancer cells and endothelial cells should be considered. Further investigation by single-cell analysis is required to determine the mechanism of the interaction between cancer cells and endothelial cells in the tumor microenvironment.

Our current work has some limitations. The PBRM1 expression was evaluated using only TMA, including the highest-grade area. Even considering intratumoral heterogeneity, however, we showed that the PBRM1 expression was correlated with the architectural patterns. Next, we semiquantitatively assessed PBRM1 IHC expression in cancer cells using an H-score. Furthermore, we could not validate the association of architectural patterns with *PBRM1* mutation status. Despite these limitations, we comprehensively showed the association of the PBRM1 expression profile with clinicopathological factors, including detailed histomorphological features.

## 5. Conclusions

We demonstrated that PBRM1 expression of cancer cells correlated with histomorphological features of ccRCC and correlated with the expression of vascular endothelial cells. Our PBRM1 expression profile indicated that PBRM1 expression in both cancer and endothelial cells may be regulated in an orchestrated manner.

## Figures and Tables

**Figure 1 cancers-14-01062-f001:**
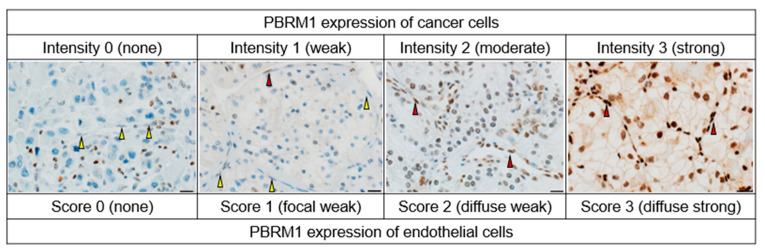
Representative PBRM1 expressions of cancer cells and endothelial cells. The staining intensity of cancer cells is assessed as follows: 0, none (internal control shows positive staining); 1, weak; 2, moderate; 3, strong. The score of endothelial cells is separately assessed as follows: 0, none; 1, focal weak; 2, diffuse weak; or 3, diffuse strong. The negative and positive expressions of endothelial cells are indicated by yellow and red arrows, respectively. Scale bar: 20 µm.

**Figure 2 cancers-14-01062-f002:**
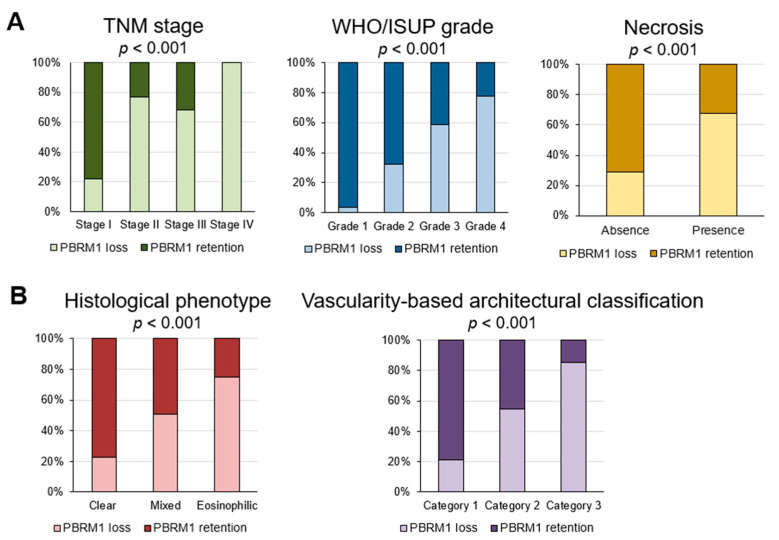
Association of PBRM1 expression in cancer cells with pathological factors. (**A**) Percentage of cases of PBRM1 expression and conventional pathological prognostic factors; (**B**) Percentage of cases of PBRM1 expression and histological phenotype and vascularity-based architectural classification.

**Figure 3 cancers-14-01062-f003:**
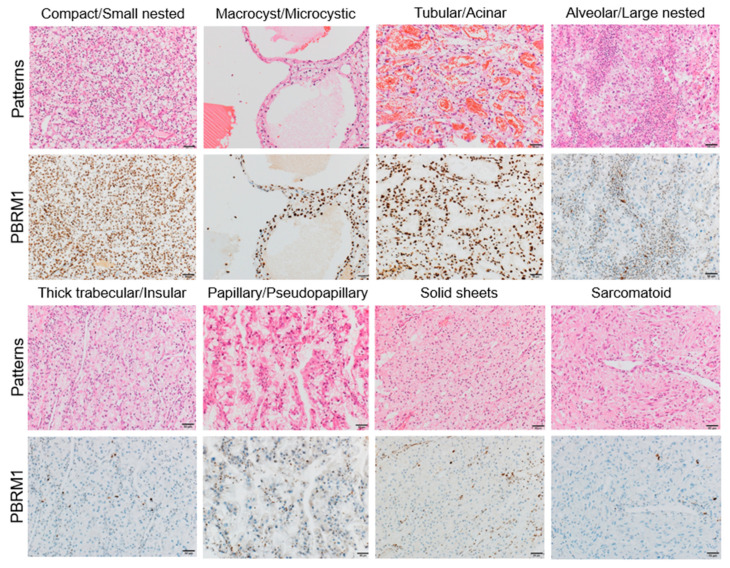
Representative images of each architectural pattern and PBRM1 immunohistochemical expression. Compact/small nested, macrocyst/microcystic, and tubular/acinar patterns are highly associated with PBRM1 retention, whereas the other patterns are highly associated with PBRM1 loss. Scale bar: 20 µm.

**Figure 4 cancers-14-01062-f004:**
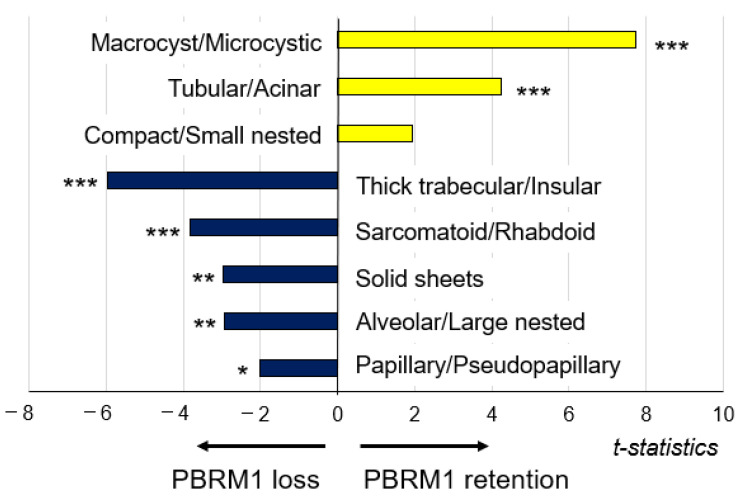
Association of architectural patterns with PBRM1 expression in cancer cells; correlation analysis between architectural patterns in the highest-grade area and PBRM1 expression (*n* = 425). * *p* < 0.05, ** *p* < 0.01, *** *p* < 0.001 using multiple linear regression analysis.

**Figure 5 cancers-14-01062-f005:**
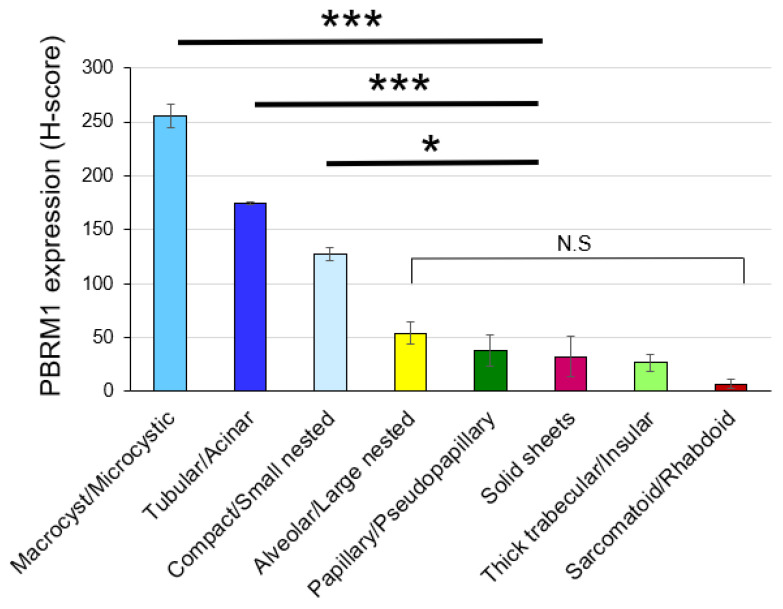
Association of architectural patterns with PBRM1 expression in cancer cells based on H-score in each TMA core (*n* = 828). The histogram shows the mean ± standard error of the mean H-score of PBRM1 expression in cancer cells. One-way analysis of variance with the Tukey test was used for statistical analysis (N.S. means not statistically significant: * *p* < 0.05, *** *p* < 0.001).

**Figure 6 cancers-14-01062-f006:**
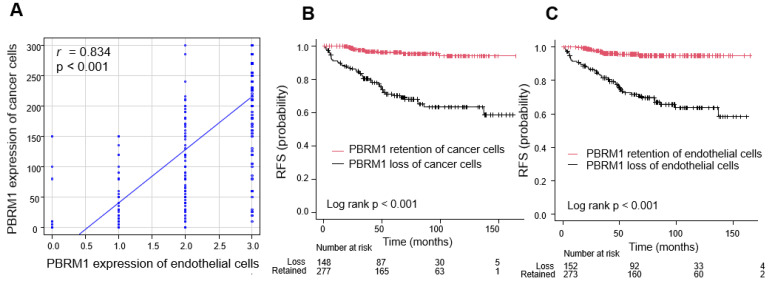
Association between cancer cells and endothelial cells. (**A**) Correlation between PBRM1 expression in cancer cells and endothelial cells. Correlations between the two variables were evaluated using Spearman’s rank correlation test. (**B**,**C**) Kaplan–Meier curve of recurrence-free survival (RFS) stratified by PBRM1 expression. (**B**) PBRM1 expression of cancer cells. (**C**) PBRM1 expression of endothelial cells.

**Table 1 cancers-14-01062-t001:** PBRM1 expression in cancer cells with clinicopathological factors in 425 cases with nonmetastatic ccRCC.

Variables	PBRM1 Retention	PBRM1 Loss
Gender, *n* (%)		
Female	81 (71.7)	32 (28.3)
Male	196 (62.8)	116 (37.2)
TNM stage, *n* (%)		
I	242 (78.1)	68 (21.9)
II	3 (23.1)	10 (76.9)
III	32 (32.0)	68 (68.0)
IV	0 (0.0)	2 (100.0)
WHO/ISUP grade, *n* (%)		
1	58 (96.7)	2 (3.3)
2	155 (68.0)	73 (32.0)
3	58 (52.7)	52 (47.3)
4	6 (22.2)	21 (77.8)
Necrosis, *n* (%)		
Absent	256 (71.1)	104 (28.9)
Present	21 (32.3)	44 (67.7)
Histological phenotype, *n* (%)		
Clear	201 (77.3)	59 (22.7)
Mixed	71 (49.0)	74 (51.0)
Eosinophilic	5 (25.0)	15 (75.0)
Vascularity-based architectural classification, *n* (%)		
Category 1	218 (79.0)	58 (21.0)
Category 2	55 (45.1)	67 (54.9)
Category 3	4 (14.8)	23 (85.2)
Recurrence, *n* (%)	11 (19.3)	46 (80.7)
Cancer-specific mortality, *n* (%)	2 (13.3)	13 (86.7)

**Table 2 cancers-14-01062-t002:** PBRM1 expression in cancer cells with histomorphological features in 425 cases with nonmetastatic ccRCC.

Architectural Patterns, *n* (%)	PBRM1 Retention	PBRM1 Loss
Compact/Small nested	127 (71.8)	50 (28.2)
Macrocyst/Microcystic	35 (97.2)	1 (2.8)
Tubular/Acinar	56 (88.9)	7 (11.1)
Alveolar/Large nested	27 (57.4)	20 (42.6)
Thick trabecular/Insular	18 (32.7)	37 (67.3)
Papillary/Pseudopapillary	10 (50.0)	10 (50.0)
Solid sheets	1 (11.1)	8 (88.9)
Sarcomatoid/Rhabdoid	3 (16.7)	15 (83.3)

## Data Availability

The data are available upon reasonable request by contacting the corresponding author.

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
