# Peer review of "PBRM1 Immunohistochemical Expression Profile Correlates with Histomorphological Features and Endothelial Expression of Tumor Vasculature for Clear Cell Renal Cell Carcinoma"

_cancers, 2022, doi:10.3390/cancers14041062_

Round 1

Reviewer 1 Report

The authors herein analyze the expression of PBRM1 in tumor cells a series of ccRCCs and its correlation with architectural patterns and PBRM1 staining of endothelial cells. This is a very interesting and well-designed study. Some minor issues need to be addressed:

  • please check the language throughout the manuscript, especially typos, such as "compact/small nested" (Figure 3)
  • the significant association between PBRM1 expression and RFS should be included in the abstract
  • the discussion section may be improved by citing and discussing other relevant works in the field, namely 10.1007/s10147-019-01564-1; 10.1016/j.urolonc.2017.10.027; 10.1371/journal.pone.0179610

Reviewer 2 Report

I consider that the resultscan be improved.  A few suggestions for improvement of the manuscript follow:

- Fig. 1

The authors should show a western blot experiment, in order to demonstrate the specificity of antibody (band of expected size etc.) used in immunohistochemistry. In my opinion, this is an important control.

Fig. 5

The standard deviation (or error the Authors should specify this point) reported in the graph of Fig. 5, appear very large values), whereas the Authors found statistically significant difference among groups. The Authors should explain the reason(s) and better describe this graph.

Reviewer 3 Report

Saiga et al. present a study demonstrating the correlation between PBRM1 immunohistochemical expression and the morphological patterns and its endothelial expression in tumor vasculature in ccRCC patients. Meanwhile, the authors conclude that PBRM1 expression is regulated in an orchestrated manner in both cancer and endothelial cells. Overall, the study showed good data quality, and the manuscript is prepared in a proper format of Cancers. The comments for the authors are listed as follows:

  1. Most of the data reflect the author’s previous findings regarding the correlation of PBRM1 in TNM stage, WHO/ISUP grade, histological phenotypes, and vascularity-based architectures (PMID: 32940805, 34580162). The new finding is that endothelial expression was tended to be lost in cases with low PBRM1 expression. Even though, the authors did not translate these findings into the clinical application. Is PBRM1 alone could represent the clinical outcomes? In what ways that the examination of PBRM1 levels could help the clinical decisions?
  2. In line 223, please specify that it is the association of PBRM1 in cancer cells and endothelial cells, as well as the title of Figure 6 in line 234.
  3. In line 224, please explain more about the data, is it positive correlated or negative correlated? And what is the purpose to examine the association of PBRM1 between cancer and endothelial cells?
  4. The authors conclude that PBRM1 expression in both cancer and endothelial cells may be regulated in an orchestrated manner. It would be nice if the authors discuss more regarding the possible mechanisms based on known function of PBRM1 that might affect the architectural patterns of tumor vasculature.

Round 2

Reviewer 2 Report

I think that the revised version of this manuscript is suitable for publication in Cancers Journal

Reviewer 3 Report

Thank to the author's response, and I have no more questions.